# Nutritional and Antioxidant Valorization of Grape Pomace from Argentinian Vino De La Costa and Italian Cabernet Wines

**DOI:** 10.3390/foods14132386

**Published:** 2025-07-05

**Authors:** Luciano M. Guardianelli, María V. Salinas, María C. Puppo, Alyssa Hidalgo, Gabriella Pasini

**Affiliations:** 1Consejo Nacional de Investigaciones Científicas y Técnicas (CONICET), Centro de Investigaciones Científicas de la Provincia de Buenos Aires (CICPBA), Facultad de Ciencias Exactas, Universidad Nacional de La Plata, CIDCA, 47 y 116, La Plata 1900, Argentina; mvsalinas@biol.unlp.edu.ar (M.V.S.); mcpuppo@quimica.unlp.edu.ar (M.C.P.); 2Facultad de Ciencias Agrarias y Forestales, Universidad Nacional de La Plata, 60 y 119, La Plata 1900, Argentina; 3Department of Food, Environmental and Nutritional Sciences (DeFENS), University of Milano, Via Celoria 2, 20133 Milan, Italy; alyssa.hidalgovidal@unimi.it; 4Department of Agronomy, Food, Natural Resources, Animals and Environment (DAFNAE), University of Padova, Viale dell’Università 16, 35020 Legnaro, Italy

**Keywords:** grape pomace, Isabella, Cabernet, chemical composition, minerals, fatty acids, polyphenols, antioxidant activity

## Abstract

Wine production generates by-products that require proper management and reuse to minimize their environmental impact. Grape pomace, a by-product of winemaking, holds significant nutritional and bioactive potential. This study evaluated the nutritional and antioxidant profiles of pomace from Isabella grapes (La Plata, Argentina) and Cabernet grapes (Veneto, Italy). Both varieties contain high levels of dietary fiber, especially Cabernet. However, Cabernet showed lower protein and lipid levels than Isabella. Calcium, potassium, and phosphorus were the major minerals in both by-products. Isabella exhibited a higher content of essential polyunsaturated fatty acids, particularly α-linoleic acid, while Cabernet shows a greater proportion of saturated and monounsaturated fatty acids. Additionally, Isabella exhibited significantly higher levels of caffeic acid derivatives (506.4 vs. 38.2 mg/kg dry weight), catechin (1613.2 vs. 294.8 mg/kg dry weight), epicatechin (1229.2 vs. 230.3 mg/kg dry weight), and total anthocyanins (2649 vs. 607.5 mg kuromanin/kg dry weight), as well as a greater total polyphenol content and antioxidant activity compared to Cabernet. These results highlight grape pomace’s potential as a valuable functional ingredient.

## 1. Introduction

The wine industry is one of the largest in the world; however, it has the disadvantage of generating various by-products and/or wastes that have a great environmental impact, since they are not well used and, thus, become environmental pollutants [1]. The winemaking process generates several by-products, including grape stems, grape pomace, wine lees, used filter cakes, stillage, and wastewater. Of these, grape pomace is the most significant, as it represents approximately 25–30% of the total waste generated by the industry. This by-product consists of stalks, skins, fragmented cells of the grape pulp, and seeds that remain after the grapes have been crushed and pressed [2]. This by-product is recognized to be a source of essential as well as non-essential nutritional components [3] with a great potential for its utilization in the food industry as ingredient for the manufacture of various novel and/or functional foods for human consumption. Its use not only contributes to waste reduction but also represents a potential source of extraordinary income.

Grape pomace contains minerals (~5%), proteins (~10–15%), and total dietary fiber (~50–60%) as well as several phenolic compounds [4,5,6,7] with antioxidant properties that make it an excellent ingredient to complement refined flours. In this sense, different authors have incorporated wine pomace into different foods, such as pasta [4], bread [5], and other bakery products [6,7]. However, there is still a lack of alternative uses with economic value, because this by-product is still undervalued, being used in distilleries or discarded in landfills. Regarding distilleries, it is traditionally used to produce spirits and liqueurs [8]. In addition, it is often used as a fertilizer, although with restrictions due to its high phenol content, which is potentially toxic to soil microbiota [9].

According to the General Wine Law, in Argentina, all the area cultivated with grapes for wine production should be of *Vitis vinifera* varieties. However, there are regions where this variety could not adapt, so immigrants, mostly Italians, have introduced the Isabella variety (a hybrid grape resulting from a random cross between the American species *Vitis labrusca* L., and an unknown variety of *V. vinifera*) from which Vino de la Costa is produced [10] and reflects both regional and cultural identity. Although it does not have a large production, it has an estimated annual output of approximately 50,000 L [11]. However, its main attraction lies in the fact that the Argentinean Institute of Viniculture recognized the wine from Isabella as a “regional wine” in 2013 [12]. Although several studies on the characterization of different pomaces are currently available [6,13,14,15,16], little is known about from that Vino de la Costa. Therefore, within the framework of Argentine–Italian collaboration, the aim of this work was to carry out an exhaustive chemical characterization and an evaluation of the antioxidant activity of grape marc from Vino de la Costa (Isabella variety), using as a reference the grape marc from Cabernet wine produced in the Veneto region (Italy), given its greater standardization and wide knowledge base in the scientific literature.

## 2. Materials and Methods

### 2.1. Materials

Red grape pomaces from two varieties were used: Italian Cabernet Sauvignon (*Vitis vinifera* L.) (kindly supplied by Acquavite S.p.A. Vazzola, di Treviso, Italy) and cv. Isabella (*Vitis labrusca* L.) resulting from the Argentinean Vino de la Costa production (provided by Cooperativa del Vino de la Costa, Berisso, Buenos Aires, Argentina).

After the winery vinification process, the must was separated from grape marc and the latter was collected. Then were dried in an oven at 65 °C until reaching the final moisture to ≤10% for both samples. Dried pomaces were ground in a laboratory mill to a size of 0.8 mm and then stored in airtight containers at 4 °C until further evaluation.

### 2.2. Methods

#### 2.2.1. Color Determination

Grape pomace flour color was determined using a colorimeter (CR-300 Konica, Minolta, Tokyo, Japan), by the measurement of L *, a *, and b * parameters on the CIELAB color scale. An L * value (whiteness) of 0 corresponds to pure black, while a value of 100 corresponds to pure white. The parameter a* value represents the intensity from green to red within the scale −60 (pure green) to +60 (pure red). The value of −60 on the b * axis is indicative of pure blue, while +60 represents a pure yellow color. The sample was poured into a Petri dish with a diameter of 5 cm, ensuring that the entire base was covered, and readings were recorded at eight different points.

#### 2.2.2. Chemical Composition

Composition analysis of the grape pomace flours was determined in triplicate according to the AOAC [17] procedures. Moisture, ash, lipid, and total protein content (using 6.25 as the nitrogen-to-protein conversion factor) were determined. The total dietary fiber (TDF) was quantified following the AOAC [17] enzyme gravimetric method (Megazyme kit). The availability of carbohydrates different to fiber was determined by difference (100-%moisture-%ash-%lipids-%protein-%TDF).

#### 2.2.3. Mineral Composition

The grape pomace flours (0.5 g) were mineralized with 7.0 mL of 67% high-purity HNO_3_ (Romil Ltd., Cambridge, UK) and 2 mL of 30% H_2_O_2_ at 200 °C for 20 min using a microwave digester (ETHOS EASY, Milestone Srl, Sorisole, Bergamo, Italy). The digested samples were diluted with ultra-pure grade water at 25 °C up to a 25 mL volume and then analyzed for mineral content.

The content of Al, B, Ba, Ca, Cr, Cu, Fe, K, Mg, Mn, Mo, Na, Ni, S, Sr, Ti, and Zn was determined in duplicate by means of an ICP-OES (inductively coupled plasma–optical emission spectrometer), namely the SPECTRO ARCOS (Spectro Analytical Instruments GmbH, Kleve, Germany). All operating parameters were optimized for a 30% nitric acid solution, as follows: plasma observation radial, nebulizer crossflow, spray chamber Scott double pass, torch injector quartz diameter 2.0 mm, plasma power 1425 W, coolant gas 12.0 L/min, auxiliary gas 0.8 L/min, nebulizer gas 0.88 L/min, sample uptake rate 2.0 mL/min, replicate read time 28 s, replicates 3, and pre-flush time 60 s. Calibration standards were matched with 1% absolute ethanol (Prolabo VWR International PBI S.r.l. Milano, Italy). The elements standards to be determined were added from single-element solutions (Inorganic Ventures, Christiansburg, VA, USA) and the concentrations of the calibration solutions were between 0 and 100 mg/L for all elements.

#### 2.2.4. Fatty Acid Profile

Lipids were extracted using the in situ transesterification method [18]. Here, 50 mg of each grape pomace sample was weighed, and 100 μL of methylene chloride and 1 mL of 0.5 N NaOH in methanol were added to them. After being flushed with nitrogen and capped, the tubes were heated at 90 °C for 10 min. The tubes were then allowed to cool slightly and 1 mL of 14% BF_3_ in methanol was added. After nitrogen flushing and capping, the heating continued again for 10 min at 90 °C. After cooling, 1 mL of MiliQ water and 500 μL of hexane were added, then homogenized and centrifuged for 10 min at 1333× *g* (Rolco, Buenos Aires, Argentina). The upper layer with the methyl esters was filtered (0.22 μm). Finally, the methyl esters (1 μL) were injected into a gas chromatograph (Agilent Technologies 7890A (Agilent Technologies; Santa Clara, CA, USA)) with flame ionization detectors. A DB 23 capillary column was utilized. The split mode was 50:1, and the oven temperature program was as follows: isotherm 1 min at 50 °C, from 50 to 175 °C at 25 °C/min, from 175 to 230 °C at 4 °C/min, and isotherm at 230 °C during 15 min. The flow rate (helium) was 0.4763 mL/min. Injector and detector temperatures were held to 250 and 280 °C, respectively. Retention times of fatty acids of samples were identified by comparison with standards of Supelco 37-Component FAME Mix fatty acid methyl esters (Sigma-Aldrich, St. Louis, MO, USA). The areas of the different peaks were analyzed using PeakFit software (version 4.12 for Windows, SPSS Inc., Chicago, IL, USA) and the results were expressed as g of fatty acid per 100 g of lipid. Assays were performed in duplicate.

#### 2.2.5. Phenolic Composition

The free phenolic compounds of grape pomace flours were extracted as described by Nakov et al. [7] with some variations. Briefly, 0.5 g of sample was extracted three times with 15 mL of 80% methanol. The pooled extracts were evaporated under a vacuum using a Laborota 4000 rotavapor (Heidolph Instruments GmbH & Co. KG, Schwabach, Germany) at 35 °C, completely dried by nitrogen flow, and resuspended in 2 mL of 80% methanol solution, filtered with a 0.45 μm PTFE membrane and analyzed by RP-HPLC. The analysis was performed using a Adamas ^®^ C18-AQ 5 μm 4.6 mm × 250 mm column and a C18 5 μm 4.6 mm × 10 mm precolumn (Sepachrom SRL, Rho, Italy) thermostated at 30 °C, as well as an L-2130 pump, an L-2300 column oven, and an L2450 diode array detector (Elite La Chrom, Hitachi, Tokyo, Japan). Gradient elution was performed using acetonitrile (A) and 1% (*v*/*v*) formic acid in water (B) mobile phases at a flow rate of 1.0 mL/min, following the gradient profile of 0–10 min from 10% to 25% A, a 10–20 min linear rise up to 60% A, and a 20 to 30 min linear rise up to 70% A, followed by a 10 min reverse to the initial 10% A with 5 min of equilibration time.

For peak quantification, calibration curves were constructed using standards from Sigma-Aldrich (St. Louis, MO, USA) and recorded at 280 nm for catechin (2.0–99.2 mg/L), epicatechin (3.2–85.0 mg/L), gallic acid (2.2–101.6 mg/L), p-coumaric acid (0.80–3.50 mg/L), p-hydroxybenzoic acid (1.05–70.4 mg/L), protocatechuic acid (0.8–27.3 mg/L), and syringic acid (1.03–10.9 mg/L); at 320 nm for caffeic acid (8.73–174.6 mg/L) and resveratrol (1.26–25.2 mg/L); and at 360 nm for kaempferol (0.1–9.5 mg/L), myricetin (1.3–17.2 mg/L), and quercetin (1.2–21.6 mg/L). Assays were performed in duplicate. Results are expressed as mg/kg of grape pomace on dry weight (d.w.).

The total anthocyanins were extracted and measured as reported by Nakov et al. [6] using a V650 spectrophotometer (Jasco, Tokyo, Japan). The results are expressed as mg/kg d.w. on the basis of the cyanidin 3-glucoside (kuromanin) standard calibration curve.

#### 2.2.6. Total Polyphenols Content and Antioxidant Activity

Extraction Process

Grape pomace samples (0.1 g) were extracted for 60 min with 1 mL of 70% methanol, under continuous stirring in an ice bath. Subsequently, the dispersion was centrifuged for 10 min at 14,000 rpm (Hettich ^®^ MIKRO 200R, Tuttlingen, Germany) and 4 °C. Determinations were performed on the supernatants in duplicate.

Determination of total phenolic compounds

Total polyphenols were determined by the Folin–Ciocalteu (FC) method according to Nakov et al. [6] with some modifications. Briefly, 1 mL of extract or extraction solvent (70% methanol) was mixed with FC reagent (0.5 mL of FC diluted 1:3) and 5 mL of a 10% *w*/*v* Na_2_CO_3_ solution prepared in a 0.1 N NaOH solution. After 30 min of incubation at room temperature, the absorbance was measured at 750 nm in a UV–visible spectrophotometer (Perkin-Elmer, Norwalk, CT, USA). A calibration curve was prepared using a solution of gallic acid in a concentration range between 0.0025 and 0.2 mg/mL. The results were expressed as mg of gallic acid equivalent (GAE)/g of grape pomace (d.w.).

Determination of antioxidant capacity

Ferric ion-Reducing Antioxidant Power Assay (FRAP)

A FRAP assay was determined according to Nakov et al. [6] with some modifications. Here, 900 μL of fresh FRAP reagent was mixed with 100 μL of the previously obtained extract, and the absorbance was measured at 593 nm after 30 min of reaction. A calibration curve containing different concentrations (0–60 μg/mL) of Trolox (6-hydroxy-2,5,7,8-tetramethylchroman-2-carboxylic acid) was used. FRAP values are presented as mg Trolox equivalent (TE)/g of grape pomace (d.w.).

Free Radical Scavenger Activity on 2,2-Diphenyl-1-Picrylhydrazyl (DPPH)

Radical scavenging activity was determined using DPPH as a free radical according to Nakov et al. [6] with some modifications. Here, 5 μL of extract or a blank solvent (70% methanol) were mixed with 2 mL of a methanol–water solution (70:30) containing 25 mg/L of DPPH. After 15 min, the absorbance was measured using a spectrophotometer at 517 nm. A standard curve was obtained using a Trolox standard solution at various concentrations (2–20 μmol/L). Antioxidant activity was expressed as mg of Trolox/g of grape pomace (d.w.).

Statistical Analysis

Data were analyzed using INFOSTAT software (Version 2024). An analysis of variance (ANOVA) was performed, and mean comparisons were carried out using the LSD (least significant difference) test at a significance level of *p* < 0.05. The results are expressed as means ± standard deviations for each parameter.

## 3. Results and Discussions

### 3.1. Grape Pomace Flours Color

Figure 1 shows the images of the flours obtained from the two grape pomaces and the respective values of the color parameters L *, a *, and b *. In the case of the Isabella, a tendency towards reddish tones (>a*) was observed, with a lighter shade (>L*) than in the case of the Cabernet. Different authors studied the color of other varieties of grape pomace. Machado et al. [19] found that the color parameters for “*Vinhão*” wine grape pomace were L * = 21.19, a *= 21.36, and b * = 0.25. On the other hand, Gerardi et al. [20], when evaluating grape pomace from “*Negroamaro*” wine, found that the color parameters were L * = 44.23, a * = 6.13, and b * = 4.89. The difference in color parameters could be attributed to the differences in the content of phenolic compounds developed by each variety. The reddish hue observed in grape pomace has been widely attributed to the presence of anthocyanins [21]. Additionally, the higher luminosity observed in Isabella grape pomace may be linked to its greater seed content. Since grape seeds contribute less pigment than the skin, their higher proportion could result in a lighter-colored flour.

### 3.2. Chemical Composition of Grape Pomace Flours

The percentage composition of the two types of grape pomace is presented in Figure 2. The moisture content in the Isabella sample was 3.7 g/100 g, similar to that found for the pomace of “*Vinhão*” wine [19] while in the Cabernet sample, it was 6.9 g/100 g, similar to pomace of “*Negroamaro*” wine [20]. Regardless, the low moisture content after dehydration indicates that the grape pomace flour will be less susceptible to microbiological spoilage reactions during storage.

A higher content of ash was observed for Cabernet (7.0 g/100 g) with respect to Isabella pomace (5.5 g/100 g), and both values were within the range reported by Beres et al. [21] for different grape varieties, ranging from 3.3 to 12.5 g/100 g, while Antonić et al. [22] reported a range of values from 1.73 to 9.10 g/100 g for other grape pomace varieties. The differences in ash content, which correspond to the mineral content of the samples, may be related to the fact that the harvest (soil, weather) and vinification conditions and practices differed between the grapes. The type and duration of the maceration process certainly affects the extraction and reabsorption of minerals during vinification, which has a significant impact on the residual mineral content in the pomace wine [23].

The protein content for Isabella and Cabernet pomaces (12.0–11.6 g/100 g) was similar to the content reported by Nakov et al. [6] (11.4 g/100 g) and within the ranges reported by García-Lomillo & González-SanJosé [8] (6–15 g/100 g) and by Antonić et al. [22] (3.6–14.2 g/100 g). The protein content also depends on the grape variety and the harvesting conditions [8].

The lipid content, mainly present in the seeds, was higher for Isabella (9.6 g/100 g), which had twice the amount found for Cabernet (4.6 g/100 g). Isabella grape pomace exhibited a higher lipid content than previously reported by Ferreira [24], Cilli et al. [25], and Nakov et al. [6], who found values of 7.3 g/100 g, 2.8 g/100 g, and 1.39 g/100 g, respectively. In contrast, the lipid content of Cabernet was consistent with the findings of Deng et al. [26] and Rainero et al. [27], who reported values of 4.7 g/100 g and 4.4 g/100 g, respectively. The differences in lipid content can be attributed to the variety, in particular to the different number of seeds present in the grape berry; considering that the main constituent of grape seeds is oil [28], it can be expected that the lipid content of Isabella grape pomace will be higher compared to other pomaces, as it is a grape with a high number of seeds per berry.

The total dietary fiber (TDF) content was significantly higher in Cabernet grape pomace (64.5 g/100 g vs. 57.5 g/100 g for Isabella). The TDF content for Isabella was higher than that found by Machado et al. [29] and Cilli et al. [25], with values of 41.9 g/100 g and 31.8 g/100 g, respectively. The TDF content for Cabernet grape pomace was higher than that found by Deng et al. [26] and Rainero et al. [27], with values of 53.2 g/100 g and 57.0 g/100 g in each case. Several researchers reported that total dietary fiber (TDF) was mainly composed of pectin (37–54 g/100 g), cellulose (27–37 g/100 g), lignin (16.8–24.2 g/100 g) and other polysaccharides [30,31]. On the other hand, Gül et al. [32] reported that the fiber content is higher in seeds than in skins; however, this trend was not observed in our results, where TDF values were lower for Isabella.

From a nutritional point of view, the grape pomace flour could be successfully used as an ingredient for the production of fiber-rich functional baked goods [33,34] to support the recommended daily intake of approximately 30 g/day for adults. In addition, grape pomace fiber is mainly represented by the insoluble dietary fiber fraction, which leads to rapid gastric emptying and promotes digestive regularity [35].

Finally, the content of available carbohydrates was not significantly different between the two types of pomaces, being 14.7 g/100 g and 12.3 g/100 g for Isabella and Cabernet, respectively. These values are within the range reported by Antonić et al. [22] for different red grape pomaces. It is expected that the content of carbohydrates other than fiber is low in red grape pomace because the whole grape is fermented in the winemaking process, unlike white and rosé wines, where the juice is fermented directly without the pomace [36].

### 3.3. Mineral Content and Fatty Acid Profile of Grape Pomace Flours

A great diversity of minerals for both grape pomaces were analyzed and are shown in Table 1. The minerals found in lower amounts (<1 mg/kg) were Mo, Ni, and Cr, with the latter being significantly higher in the Cabernet variety. In addition, 2.7 times more Ba was found in Isabella than in Cabernet. In both pomaces, similar amounts of Al were found (≈33 mg/kg). Compared to the Cabernet variety, Isabella contained a lower content of Mn, Zn, and B but higher levels of Sr and Na. Some of the minerals studied were found to be within the range reported in the literature by various authors [22]. Nickel (Ni), chromium (Cr), and aluminum (Al) are elements that should be consumed in low doses, given that they, especially Al, tend to accumulate in the organism and have neurotoxic effects. However, the levels of these metals found in the pomace of Isabella and Cabernet varieties were lower than those reported by Pereira et al. [15]. Pereira reported, for Cabernet Sauvignon (Cs3), values of 2.5, 3.75, and 137 mg/kg pomace for Cr, Ni, and Al, respectively, while we obtained values (mg/kg) for Isabella and Cabernet of 0.36/0.55 for Cr, 0.43/0.39 for Ni, and 33.9/32.4 for Al. While the tolerable daily intake (TDI) for Cr (III) is 0.3 mg/kg w.b. and daily dietary Al intakes are estimated at 8–10 mg/day [37,38], it is important to note that Cr (III) intake should, ideally, be 18–21 mg/day for adults. Nevertheless, if these pomaces are incorporated into foods, like breads and cakes, their proportion should not exceed 10% due to their negative impact on gluten development and the overall sensory attributes of the final product.

Other minerals found in both pomaces in amounts around 100 mg/kg are Fe and Cu, with the latter being significantly higher in the Isabella variety (Table 1). High amounts of S, Mg, and P were also detected, with Mg and P in high proportions in the Argentinean grape pomace.

Ca and K were the more abundant minerals found in both varieties. In particular, the K ranged from 20,482 (for Isabella) to 25,510 mg/kg (for Cabernet), which represents approximately 80% of all cations in grape pomace, as this mineral is primarily found in grape skins [39]. It is known that K can contribute to the metabolism and synthesis of proteins and glycogen, can regulate the water content in the organism [40], and is considered essential in maintaining the osmotic balance in humans [41]. This mineral also plays an important role in reducing blood pressure and the risk of osteoporosis by reducing urinary calcium excretion [42]. Nevertheless, the potassium content in foods or drinks containing pomace flour should be considered. To have a positive impact on human health, a minimum intake of 2600 mg/day for women and 3400 mg/day for men is recommended. Regarding the forms in which this K can be found, Nurgel and Canbas [43] reported that tartrates are the most abundant potassium salts, with significant variations depending on the ripening stage and cultivation practices used for wine grapes.

Finally, the Ca content ranged from 6017 ± 114 mg/kg (d.w.) in Cabernet to 4197 ± 84 mg/kg (d.w.) in Isabella (Table 1). In fortified foods or beverages, its contribution depends on the percentage of substitution. Only at levels of 1000 mg/day for men and 1200 mg/day for women can it support bone and tooth health, aid muscle contraction and relaxation, and play a role in nerve function, blood pressure regulation, and blood clotting [15]. Based on these analyses, Cabernet grape pomace would constitute a more suitable food ingredient than Isabella due to its higher content of minerals, such as Mn, Zn, Fe, S, Ca, and K.

The fatty acid profiles of both grape pomace flours are shown in Table 1. Both pomaces have fatty acids ranging from 8 to 20 carbon atoms. Approximately 90% of the total fatty acids are linoleic acid (18:2 n6), followed by oleic acid (18:1 n9) and palmitic acid (16:0). Although these three fatty acids are found in both pomaces in the highest proportions, the essential linolenic acid (C18:3 n3) content is significantly higher in the Isabella grape, while oleic and palmitic acid are found to a greater extent in the Cabernet grape (Table 1). The high content of linolenic acid in Isabella grape pomace could be due to the presence of grape seeds. Several authors have reported that this essential fatty acid is mainly found in the seeds [21]. The total unsaturated fatty acids are higher for the Isabella grape pomace than for the Cabernet variety. Several authors found similar results when analyzing different grape varieties, finding that the percentages of fatty acids were comparable to those observed in this study. However, the variability can be attributed to the different grape varieties [21]. In Table 1, the PUFA/SFA ratio, atherogenic index (AI), and thrombogenic index (TI) were calculated according to the method proposed by Ulbricht and Southgate [44]. The Isabella variety showed a higher PUFA/SFA ratio and lower AI and TI values compared to Cabernet. This trend suggests that the Isabella variety has a better lipid profile than Cabernet, which is associated with a lower risk of cardiovascular disease.

### 3.4. Phenolic Compounds and Antioxidant Activity of Grape Pomace Flours

Grapes are rich in phenolics; however, during the winemaking process, only about 30% of these compounds are extracted [45]. As a result, grape pomace remains an excellent source of phenolic compounds.

Phenolic compounds are secondary plant metabolites characterized by having at least one aromatic ring with one or more hydroxyl groups [46]. Grapes contain a variety of phenolic compounds; among them, the skins and seeds stand out for their tannins (proanthocyanidins), which are responsible for the astringency. Red grapes contain anthocyanins, the pigments that give them their characteristic color. Samples also contain phenolic acids, such as hydroxybenzoic and hydroxycinnamic acids, and other flavonoids, such as catechins and flavonols. The composition and concentration of phenolic compounds in wine by-products vary depending on the grape variety, climatic conditions, and the extraction method used [1,45]. The content of free phenolics in both types of pomace is shown in Table 2.

The phenolic acids in Isabella grapes were gallic acid and derivatives of caffeic and *p*-coumaric acid, with the caffeic acid derivative being the most abundant. In the Cabernet grape pomace sample, gallic acid and caffeic acid derivatives were found, with the former being the major component. Iora et al. [47] reported similar levels of gallic acid in Brazilian Cabernet pomace and Italian Cabernet (128.8 mg/kg vs. 130.7 mg/kg, respectively). Additionally, the Isabella variety presented comparable gallic acid content to Brazilian Merlot and Tanat cultivars, with values of 102.2 and 111.1 mg/kg, respectively [47]. The total content of free phenolic acids reported by Nakov et al. [6] was 126.1 ± 10.9 mg/kg in grape pomace from Muscat Hamburg grapes, which was slightly higher than that in Cabernet pomace and significantly lower than that in Isabella pomace.

Flavonoids, predominantly found in grape seeds—where they can represent between 56% and 65% of total flavonoids—are typically colorless or yellow. Red grape varieties tend to produce winemaking by-products with higher flavonol concentrations than white cultivars [45]. As for flavonoids, those found in Isabella grapes were catechin, catechin derivatives, epicatechin, myricetin, quercetin, kaempferol, and kaempferol derivatives, with catechin being the one found in the greatest quantity, while in Cabernet grapes, although the flavonoids found were very similar (with the exception of myricetin, which was not found in detectable quantities), in this case the major flavonoid detected was the catechin derivative. The higher flavonoid content in Isabella could be due to the fact that these compounds are mainly found in the seeds [45] and, given that, as previously mentioned, the Isabella grape has, on average, a higher number of seeds per grape, this would result in a higher flavonoid content. On the other hand, stilbene levels differ among grape cultivars, with red varieties generally containing higher amounts. These variations are influenced by both biotic and abiotic factors, as well as the processing techniques applied [45]. The stilbenoid content given by resveratrol was higher in the case of Cabernet. Iora et al. [47] report 5.8 and 6.2 mg/kg of this compound in Brazilian Merlot and Cabernet, respectively—values that are lower than those observed in our study. The total flavonoid content in Cabernet was similar to that reported by Nakov et al. [6] (1461.5 ± 66.2 mg/kg), while that of Isabella was significantly higher. Nakov et al. [6] did not detect resveratrol in grape pomace. The total phenolic content was higher in the Isabella grape (4020 vs. 1733 mg/kg d.w.). Finally, Table 2 shows that the total anthocyanin content in Isabella grapes was twice that found in Cabernet grapes. Therefore, Isabella grape pomace had the highest content of total polyphenolic compounds as well as of the various polyphenol subgroups (phenolic acids, flavonoids, and anthocyanins). Moreover, in the Isabella variety, the high anthocyanin content could be responsible for the color of the pomace (Figure 1).

The differences found between the two grape pomaces could be attributed to several factors, including edaphoclimatic factors, grape variety and type, type of tissue used (skins or seeds), winemaking conditions (contact or not with skins during the fermentation process) and the health status of the grapes at the time of harvest [45].

Regarding the health benefits of phenolic compounds for consumers, there is evidence that they maintain gut health by preventing chronic diseases and cancer [48,49]. In addition, the antioxidant potential of these compounds has been shown to aid in food preservation by inhibiting lipid oxidation and demonstrating antimicrobial effects [50].

The total polyphenolic content (TPC), measured by the Folin–Ciocalteu method, and the antioxidant activity (FRAP and DPPH methods) of both pomace flours are presented in Figure 3. Isabella grape pomace exhibited a higher TPC than Cabernet (Figure 3a). The values obtained for both grape pomaces were higher than the 4.92 mg GAE/g found by Abdelhakam et al. [51] in red grape pomace from Egypt, namely the Red Globel variety, and the 6.71 mg GAE/g observed in dehydrated Isabella grapes [52]. On the other hand, the content of total phenolic compounds in Isabella and Cabernet grape pomace was lower than that reported by Rockenbach et al. [16] when studying different red grape pomace varieties. This difference could be due to the fact that the content of phenolic compounds depends on several factors, including the grape variety, the geographical location, the type of climate, and the winemaking process [21]. Antioxidant activity, assessed by FRAP (Figure 3b) and DPPH (Figure 3c) methods, was higher in the Isabella sample. The values obtained for both pomaces were higher than the 3.76 mg Trolox/g measured by DPPH in Cabernet grape pomace [53]. On the other hand, they were lower than those reported by Monteiro et al. [14] and Rockenbach et al. [16] when evaluating Brazilian red and rosé wine grape pomaces.

The variation in phenolic composition between Isabella and Cabernet highlights their unique advantages for different industrial applications, reinforcing the importance of characterizing wine by-products to optimize their use in functional and sustainable product development.

## 4. Conclusions

The characterization of grape pomace flours from Isabella and Cabernet varieties revealed significant differences in their physical and chemical properties, suggesting specific applications depending on the target industry. The color of the grape pomace flour was influenced by the grape variety and its anthocyanin content, with Isabella exhibiting redder tones and higher brightness, possibly due to its higher proportion of seeds. Regarding chemical composition, differences were observed in moisture, ash content, proteins, lipids, and dietary fiber. Cabernet had a higher mineral content, with elevated contents of Fe, Zn, Ca, and K, making it a promising alternative for enriching food products with these essential minerals. On the other hand, Isabella exhibited a higher lipid content, with a greater proportion of unsaturated fatty acids, particularly linolenic acid, which could make it more attractive for applications in the food and cosmetic industries. Dietary fiber content was higher in Cabernet, reinforcing its potential use as an ingredient in fiber-rich functional products. Additionally, the low amount of available carbohydrates in both pomaces confirms their potential as ingredients in formulations with a low glycemic impact. From the perspective of bioactive compounds, Isabella grape pomace flour stood out for its high polyphenol content, especially flavonoids and anthocyanins, as well as its greater antioxidant activity, positioning it as a valuable ingredient in the food and pharmaceutical industries. Meanwhile, Cabernet showed a higher presence of stilbenes, such as resveratrol, with potential health benefits. In the context of the growing wine industry and the focus on a circular economy, these results highlight the importance of characterizing wine by-products due to the possibility of their utilization as ingredients in several industrial applications.

## Figures and Tables

**Figure 1 foods-14-02386-f001:**
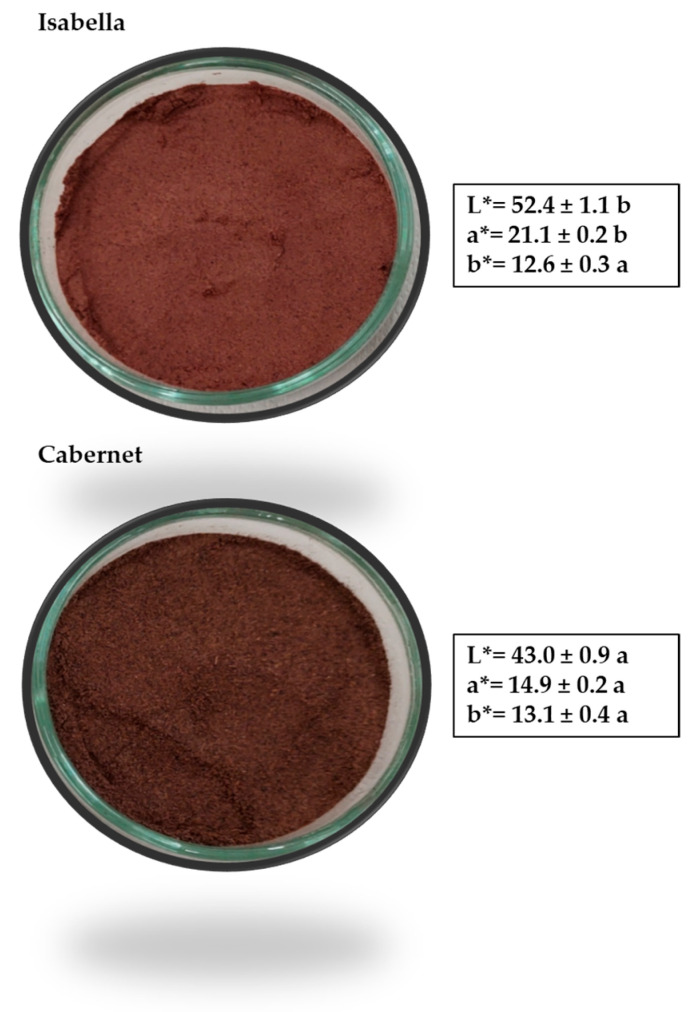
Images of the flours of Isabella and Cabernet grape pomace. CIELAB parameters (L *, a *, and b *) are included in the image. Different letters in the same parameter indicate significant differences (*p* < 0.05).

**Figure 2 foods-14-02386-f002:**
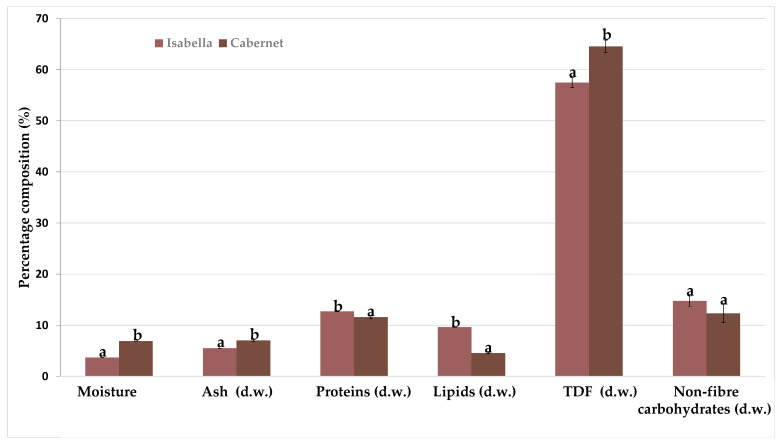
Proximal composition of Isabella and Cabernet grape pomaces. Different letters in the same compound indicate significant differences (*p* < 0.05). Values are expressed as mean ± standard deviations. TDF: total dietary fiber.

**Figure 3 foods-14-02386-f003:**
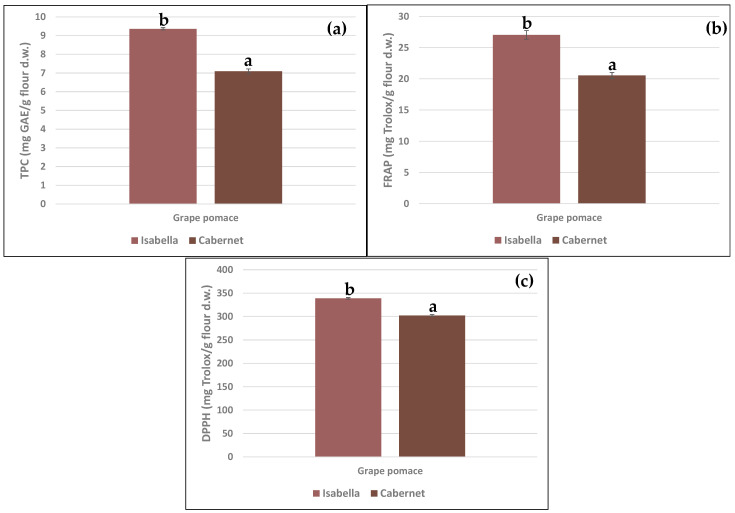
Total polyphenol content (TPC, mg GAE/g flour d.w.) by Folin–Ciocalteu (**a**) and antioxidant capacity (mg Trolox equivalent/g flour d.w.) by FRAP (**b**) and DPPH (**c**) methods in Isabella and Cabernet grape pomaces. Values are expressed as mean ± standard deviations. Different letters in the same Figure indicate significant differences (*p* < 0.05) between the two varieties.

**Table 1 foods-14-02386-t001:** Mineral content (mg/kg dry weight) and fatty acid composition (g fatty acid/100 g lipids) of Isabella and Cabernet grape pomaces.

Compounds	Grape Varieties	
	Isabella	Cabernet
Minerals		
Molybdenum (Mo)	0.27 ± 0.00 a	0.21 ± 0.03 a
Nickel (Ni)	0.43 ± 0.1 a	0.39 ± 0.00 a
Chrome (Cr)	0.36 ± 0.006 a	0.55 ± 0.02 b
Titanium (Ti)	1.91 ± 0.2 b	1.71 ± 0.03 a
Barium (Ba)	4.94 ± 0.27 b	1.84 ± 0.21 a
Aluminum (Al)	33.88 ± 1.75 a	32.40 ± 0.79 a
Manganese (Mn)	10.7 ± 0.2 a	13.59 ± 0.53 b
Zinc (Zn)	12.82 ± 0.03 a	45.71 ± 8.60 b
Boron (B)	24.85 ± 0.35 a	33.70 ± 1.54 b
Strontium (Sr)	21.3 ± 0.5 b	7.50 ± 0.31 a
Sodium (Na)	93.0 ± 0.5 b	34.47 ± 2.16 a
Iron (Fe)	77.66 ± 3.96 a	109.02 ± 20.04 a
Copper (Cu)	121.43 ± 2.7 b	86.95 ± 5.84 a
Sulfur (S)	1602 ± 23 a	1963 ± 94 b
Magnesium (Mg)	923 ± 15 b	841 ± 26 a
Phosphorus (P)	2886 ± 61 b	2448 ± 96 a
Calcium (Ca)	4197 ± 84 a	6017 ± 114 b
Potassium (K)	20,482 ± 466 a	25,510 ± 1585 b
Fatty acids		
Caprylic acid (C8:0)	0.05 ± 0.00 a	0.14 ± 0.01 b
Capric acid (C10:0)	0.16 ± 0.00 b	0.11 ± 0.00 a
Lauric acid (C12:0)	0.26 ± 0.01 a	0.28 ± 0.00 b
Myristic acid (C14:0)	0.33 ± 0.00 a	0.66 ± 0.01 b
Palmitic acid (C16:0)	12.54 ± 0.14 a	21.35 ± 0.42 b
Palmitoleic acid (C16:1)	0.16 ± 0.00 a	1.87 ± 0.01 b
Stearic acid (C18:0)	3.33 ± 0.04 a	5.38 ± 0.04 b
Oleic acid (C18:1 n9)	18.17 ± 0.01 a	19.73 ± 0.07 b
Linoleic acid (C18:2 n6)	61.13 ± 0.01 b	46.47 ± 0.30 a
α-linolenic acid (C18:3 n3)	3.15 ± 0.03 b	2.60 ± 0.08 a
Eicosanoic acid (C20:0)	0.52 ± 0.04 a	1.22 ± 0.08 b
Eicosenoic acid (C20:1 n9)	0.09 ± 0.01 a	0.09 ± 0.01 a
SFA	17.18 ± 0.15 a	29.13 ± 0.46 b
MUFA	18.42 ± 0.01 a	21.69 ± 0.06 b
n3-PUFA	3.15 ± 0.03 b	2.60 ± 0.08 a
n6-PUFA	61.13 ± 0.01 b	46.47 ± 0.30 a
PUFA/SFA	3.74 ± 0.1 b	1.68 ± 0.4 a
AI	0.17	0.34
TI	0.33	0.64

Values are expressed as mean ± standard deviations. Different letters (lower case) in the same row indicate significant differences (*p* < 0.05) between the two varieties. SFA: saturated fatty acids, PUFA: polyunsaturated fatty acids, PUFA/SFA ratio, AI: atherogenic index, and TI: thrombogenic index.

**Table 2 foods-14-02386-t002:** Phenolic compounds (mg/kg dry weight, d.w.) and total anthocyanins (mg kuromanin/kg d.w.) in Isabella and Cabernet grape pomace.

Compounds	Grape Varieties	
	Isabella	Cabernet
*Phenolic acids*		
Gallic acid	75.6 ± 0.8 a	130.7 ± 1.4 b
Caffeic acid derivatives	506.4 ± 2.8 b	38.2 ± 1.4 a
*p*-coumaric acid derivatives	11.9 ± 0.6	-
Total	594 ± 2.5 b	168.9 ± 1.4 a
*Flavonoids*		
Catechin	1613.2 ± 4.2 b	294.8 ± 1.4 a
Catechin derivatives	295.2 ± 4.2 a	786.0 ± 2.1 b
Epicatechin	1229.2 ± 2.8 b	230.3 ± 2.8 a
Myricetin	35.9 ± 1.4 B	-
Quercetin	171.6 ± 2.8 b	93.1 ± 1.4 a
Kaempferol	21.2 ± 0.7 b	2.2 ± 0.0 a
Kaempferol derivatives	21.9 ± 0.6 a	56.0 ± 1.4 b
Total	3388.3 ± 7.3 b	1462.3 ± 4.3 a
*Stilbenoids*		
Resveratrol	37.8 ± 0.6 a	102.1 ± 2.6 b
Total anthocyanins	2649 ± 114.8 b	607.5 ± 16.2 a

Values are expressed as mean ± standard deviations. Different letters (lower case) in the same row indicate significant differences (*p* < 0.05) between the two varieties.

## Data Availability

Dataset available on request from the authors.

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
