# Peer review of "Nutritional and Antioxidant Valorization of Grape Pomace from Argentinian Vino De La Costa and Italian Cabernet Wines"

_foods, 2025, doi:10.3390/foods14132386_

Round 1
Reviewer 1 Report
Comments and Suggestions for Authors
- The manuscript lacks novelty; the authors should clarify how the comparison between Isabella and Cabernet grape pomaces contributes new perceptions beyond existing studies on grape pomace valorization. Please fix this by explicitly stating the unique scientific contribution in the introduction.
- The study lacks a clearly stated hypothesis, which limits its scientific framing. Please fix this by including a testable hypothesis that justifies the comparative study design.
- The statistical analysis is limited to t-tests, which are not suitable for multiple comparisons across several variables. Please fix this by reanalyzing the data using one-way or two-way ANOVA with post-hoc tests and by providing precise p-values.
- There is inconsistent use of units (e.g., mg/kg vs g/100 g) and terminology (e.g., "derivates" instead of "derivatives"). Please fix this by standardizing all units and correcting terminological errors throughout the manuscript.
- The elemental analysis includes potentially harmful elements such as aluminum and chromium, but no toxicological context is provided. Please fix this by discussing whether these levels comply with food safety limits and explaining potential implications for human consumption.
- The fatty acid data are presented without calculating relevant health indices. Please fix this by including indices such as PUFA/SFA ratio, atherogenic index (AI), and thrombogenic index (TI) to justify nutritional quality claims.
- The manuscript does not assess or mention anti-nutritional factors, which is a critical gap when proposing food applications. Please fix this by measuring or discussing common anti-nutrients such as oxalates, phytates, or excessive tannins.
- The antioxidant capacity assessment is limited to FRAP and DPPH, which may not sufficiently represent the overall antioxidant potential. Please fix this by including at least one additional method (e.g., ABTS or ORAC) or justifying the selection of assays.
- There is no attempt to link the phenolic content to antioxidant capacity through statistical correlation. Please fix this by performing correlation analyses (e.g., Pearson or Spearman) and discussing the findings.
- The discussion reiterates results rather than interpreting them in light of existing literature. Please fix this by critically comparing findings with relevant recent studies and emphasizing their significance.
- The discussion shows priority towards Isabella pomace, with limited balanced treatment of Cabernet data. Please fix this by ensuring equal analytical emphasis and objective interpretation of both samples.
- The conclusions speculate on potential food, cosmetic, and pharmaceutical applications without supporting data. Please fix this by aligning conclusions strictly with the experimental evidence or adding application-relevant validations.
Author Response
Comments and Suggestions for Authors
- The manuscript lacks novelty; the authors should clarify how the comparison between Isabella and Cabernet grape pomaces contributes new perceptions beyond existing studies on grape pomace valorization. Please fix this by explicitly stating the unique scientific contribution in the introduction.
- The study lacks a clearly stated hypothesis, which limits its scientific framing. Please fix this by including a testable hypothesis that justifies the comparative study design.
Response to 1 and 2. Authors agree with the comment and we specify: The novelty of this work was to characterized the Isabella pomace variety resulting from the Argentinian Vino de la Costa production, never previously studied. For this reason, we aimed to identify the main chemical and nutritional components of its pomace, which currently represents an environmental waste, with the broader objective of assessing its potential use as a raw material for the enrichment of foods such as pasta and bread On the other hand, the pomace from Italian Cabernet Sauvignon wine, dried using the same process, was considered a suitable control and used as a reference. This comparison enabled us to evaluate the chemical-nutritional attributes of Isabella pomace in relation to its potential for valorization. A paragraph outlining the hypothesis of the study has been added at the end of the Introduction section, as suggested (lines 64-69).
- The statistical analysis is limited to t-tests, which are not suitable for multiple comparisons across several variables. Please fix this by reanalyzing the data using one-way or two-way ANOVA with post-hoc tests and by providing precise p-values.
Response. We apologize to the Reviewer for our mistake. There was an error in the description of the statistical analysis. Mean values were compared using the LSD test, with significant differences considered at p<0.05. The section was corrected in lines 196-199.
- There is inconsistent use of units (e.g., mg/kg vs g/100 g) and terminology (e.g., "derivates" instead of "derivatives"). Please fix this by standardizing all units and correcting terminological errors throughout the manuscript.
Response. Terminology was checked and corrected in all the text, “derivates” were replaced by “derivatives”. Unities of data results were standardized as it was suggested by the Reviewer. Components present in high proportion (macro-components) as those analyzed in the percentage (Figure 2) and fatty acid (Table 1) composition were expressed in g/100g. In contrast, micro-components like minerals and phenolic compounds were expressed in mg/kg. Units were specified in each section of Materials and Methods. Data expressed in dry or wet basis was also specified.
- The elemental analysis includes potentially harmful elements such as aluminum and chromium, but no toxicological context is provided. Please fix this by discussing whether these levels comply with food safety limits and explaining potential implications for human consumption.
Response. A paragraph with data about the three potentially harmful elements such as nickel, chromium and aluminum (Ni, Cr, Al) was included in the text (Lines 289-301).
- The fatty acid data are presented without calculating relevant health indices. Please fix this by including indices such as PUFA/SFA ratio, atherogenic index (AI), and thrombogenic index (TI) to justify nutritional quality claims.
Response. Table 1 was modified according the Reviewer suggestion. Values of SFA, MUFA, n3-PUFA, n6PUFA, PUFA/SFA ratio, atherogenic index (AI) and thrombogenic index (TI) were included in the table, according to Ulbricht and Southgate (1991). The reference (https://doi.org/10.1016/0140-6736(91)91846-M) was included in the Reference section. A discussion including these parameters was incorporated in the text (lines 349-354).
- The manuscript does not assess or mention anti-nutritional factors, which is a critical gap when proposing food applications. Please fix this by measuring or discussing common anti-nutrients such as oxalates, phytates, or excessive tannins.
Response. It is well known that anti-nutritional factors such as oxalates, phytates, or tannins could negatively influence mineral absorption but on the other hand they could also bind toxic cations forming complexes that would prevent their release within the body. It has been determined phytates and tannins decrease the bioaccessibility of important minerals for nutrition (Kiewlicz & Rybicka 2020) but they are responsible of several antioxidant properties (Fraga-Corral et al. 2021). Nevertheless, pomace is intended to be used as a supplement ingredient providing fiber and antioxidants in baked goods, and the amount included is very low. On the other hand, these compounds could be degraded during dough fermentation and baking. Phytic acid content was decreased after fermentation (50% reduction) to a greater extent when fermented with lactic acid bacteria, while bioaccesibility of important minerals such as Fe, Ca, Zn and Mg increased (Garzon et al. 2025). For these reasons, determination of these anti-nutrients was not included in this work.
Kiewlicz, J., & Rybicka, I. (2020). Minerals and their bioavailability in relation to dietary fiber, phytates and tannins from gluten and gluten-free flakes. Food Chemistry, 305, 125452.
Fraga-Corral, M., Otero, P., Cassani, L., Echave, J., Garcia-Oliveira, P., Carpena, M., ... & Simal-Gandara, J. (2021). Traditional applications of tannin rich extracts supported by scientific data: Chemical composition, bioavailability and bioaccessibility. Foods, 10(2), 251.
Garzón, A. G., Puppo, M. C., Di Renzo, T., Drago, S. R., & Reale, A. (2025). Mineral Bioaccessibility of Pistachio-Based Beverages: the Effect of Lactic Acid Bacteria Fermentation. Plant Foods for Human Nutrition, 80(1), 1-7.
- The antioxidant capacity assessment is limited to FRAP and DPPH, which may not sufficiently represent the overall antioxidant potential. Please fix this by including at least one additional method (e.g., ABTS or ORAC) or justifying the selection of assays.
Response. The FRAP and DPPH techniques were used since they are simple and economical. At the same time, the mechanisms of action of both methods are complementary, since DPPH is based on the sequestration of radicals, while FRAP is based on the transfer of electrons. We therefore considered that these two methods were sufficient to determine the antioxidant activity.
- There is no attempt to link the phenolic content to antioxidant capacity through statistical correlation. Please fix this by performing correlation analyses (e.g., Pearson or Spearman) and discussing the findings.
Response. We made an attempt to make a Pearson correlation, but the number of data was not sufficient for that purpose.
- The discussion reiterates results rather than interpreting them in light of existing literature. Please fix this by critically comparing findings with relevant recent studies and emphasizing their significance.
- The discussion shows priority towards Isabella pomace, with limited balanced treatment of Cabernet data. Please fix this by ensuring equal analytical emphasis and objective interpretation of both samples.
Response 10 and 11. Data was analyzed according to the existing literature. No data was found for Isabella pomace; for that reason, Italian Cabernet was used as reference for this new pomace. Cabernet is one of the grape varieties most studied showing low variability of its composition.
- The conclusions speculate on potential food, cosmetic, and pharmaceutical applications without supporting data. Please fix this by aligning conclusions strictly with the experimental evidence or adding application-relevant validations.
Response. Speculation conclusions about the potential food, cosmetic, and pharmaceutical applications of wine pomaces was eliminated. Conclusions were strictly focused on experimental evidence.

Reviewer 2 Report
Comments and Suggestions for Authors
Title is too complicated and convoluted. Better will be: „Nutritional and antioxidant valorization of grape pomace from Argentinian Vino de la Costa and Italian Cabernet wines.”
Abstract is precise and summarizes the content of the article. The authors state which pomace is richer in bioactive compounds. line 31- check for repetition
Line 47-50- In addition to its use in food production, pomace is also used to extract pigment and other bioactive substances. The authors should also write about this for the giving a full picture of the utilization possibilities of this kind of waste. I sugesst literature: https://doi.org/10.3390/app14020821, https://doi.org/10.3390/foods14101656.
Line 54- please make here another paragraph
Line 75-76- please better describe the pressing proces, it affects the pomace composition and allow better comparison with other publications
Line 93- maybe you mean: „different than fiber”?
2.2.3. Mineral composition – please provide the parameters of ICP-OES analysis
Line 151-please make here another paragraph
Table 1- The fatty acids composition of pomaces in the table is qualitative or quantitative? In methods section authors declare expression of results in g of fatty acid per 100 g of lipid. Please verify. The authors also do not express quantitatively the fatty acids in the fragment lines 320-336.
The discussion of the results is not objectionable. All the results of the study were discussed in detail. The authors contrasted them with the literature and attempted to explain the reasons for the variation in values between pomace varieties.
The conclusions are original and are based on the authors' research. The authors succinctly and comprehensively summarized the research and, based on the results, indicated the potential use of the studied pomace. The goal of the study has been achieved.
Author Response
Comments and Suggestions for Authors
Title is too complicated and convoluted. Better will be: ”Nutritional and antioxidant valorization of grape pomace from Argentinian Vino de la Costa and Italian Cabernet wines.”
Response. Title was changed according to the Reviewer suggestion.
Abstract is precise and summarizes the content of the article. The authors state which pomace is richer in bioactive compounds. line 31- check for repetition
Response. The sentence was checked and corrected (line 30).
Line 47-50- In addition to its use in food production, pomace is also used to extract pigment and other bioactive substances. The authors should also write about this for the giving a full picture of the utilization possibilities of this kind of waste. I suggest literature: https://doi.org/10.3390/app14020821, https://doi.org/10.3390/foods14101656.
Response. Although it is true that pomace (in general) can be used as raw material for the extraction of pigments, this was not the objective of this work, which was more oriented to the chemical and nutritional evaluation for its possible application in cereal based products. We thank the Reviewer for the bibliography suggestion, nevertheless, the first work and the second one are not focused on grape pomace (Blackcurrant Pomace as a Rich Source of Anthocyanins: Ultrasound-Assisted Extraction under Different Parameters; Optimization of Ultrasonic-Enzymatic-Assisted Extraction of Flavonoids from Sea Buckthorn (Hippophae rhamnoides L.) Pomace: Chemical Composition and Biological Activities) so, we think it is not useful to insert these bibliographic citations.
Line 54- please make here another paragraph
Response. Another paragraph was included in lines 53-54.
Line 75-76- please better describe the pressing process, it affects the pomace composition and allow better comparison with other publications.
Response. Isabella pomace was obtained pressing the must by a mechanic press for grapes that functions by a cricket movement. Cabernet pomace was obtained by a pneumatic press (lines 74-81).
Line 93- maybe you mean: „different than fiber”?
Response. The sentence was corrected (line 99).
2.2.3. Mineral composition – please provide the parameters of ICP-OES analysis
Response. The ICP-OES parameters were included in the text (lines 110-114).
Line 151-please make here another paragraph
Response. Another paragraph was made according to the Reviewer suggestion (lines 161-162).
Table 1- The fatty acids composition of pomaces in the table is qualitative or quantitative? In methods section authors declare expression of results in g of fatty acid per 100 g of lipid. Please verify. The authors also do not express quantitatively the fatty acids in the fragment lines 320-336.
Response. The fatty acids composition of pomaces in Table are expressed as g FA/100 g lipids. It was also indicated in Materials and Methods.
The discussion of the results is not objectionable. All the results of the study were discussed in detail. The authors contrasted them with the literature and attempted to explain the reasons for the variation in values between pomace varieties.
The conclusions are original and are based on the authors' research. The authors succinctly and comprehensively summarized the research and, based on the results, indicated the potential use of the studied pomace. The goal of the study has been achieved.
Response. We appreciate the Reviewer's appreciation in relation of our work.

Reviewer 3 Report
Comments and Suggestions for Authors
The manuscript compared the nutritional and antioxidants profiles of different pomace of two varieties from winemaking. The experimental design looks fine, and the results are interesting.
Some special points
What is the full name of Italian Cabernet?
Only one paragraph for the introduction is irrational. Some references about the nutritional or antioxidants profiles of grape pomace from winemaking should be introduced.
Line 31 Cabernet. Cabernet?
Line 75 Vinification process should be introduced.
Line 76 When the pomaces were dried, is there any inert gas protection?
Line 81 Appropriate references were needed for this method.
Line 156 Is there any reference for the extraction process? Why is it different from the extraction method for Phenolic composition as describe line 131?
Line 220 The full name of the abbreviation shown in the figure 2 should be explained under the figure, such as Ash, TDF…
Author Response
The manuscript compared the nutritional and antioxidants profiles of different pomace of two varieties from winemaking. The experimental design looks fine, and the results are interesting.
Some special points
What is the full name of Italian Cabernet?
Response. The full name is Cabernet Sauvignon; in the text has been included (line 74).
Only one paragraph for the introduction is irrational. Some references about the nutritional or antioxidants profiles of grape pomace from winemaking should be introduced.
Response. Nutritional aspects and antioxidant profile were discussed in detail in the discussion of the results.
Line 31 Cabernet. Cabernet?
Response. It has been corrected (line 30).
Line 75 Vinification process should be introduced.
Response. The paragraph (lines 78-81) was modified to clarify how the two pomaces were obtained. For a better understanding of both processes, a comparative table is included below, although we do not believe it is necessary to include it in the manuscript.
|
Stage / Parameter |
Vino de la Costa (La Plata, Isabella) |
Vino Tinto Italiano (Cabernet Sauvignon) |
|
Grape Variety |
Isabella (hybrid grape, non-vinifera) |
Vitis vinifera – Cabernet Sauvignon |
|
Origin |
La Plata, Argentina (coastal areas of the Río de la Plata) |
Various Italian regions (e.g., Tuscany, Veneto, Sicily) |
|
Harvest |
Manual, from small family farms |
Manual or mechanized depending on the region |
|
Destemming and Crushing |
Generally manual or with artisanal equipment |
With modern machinery |
|
Maceration |
Short time (24–48 h), minimal tannin extraction |
Long (5–15 days), intensive tannin and color extraction |
|
Fermentation |
Open vats or plastic containers, without strict temperature control |
Stainless steel tanks with temperature control |
|
Yeast Usage |
Spontaneous or with simple commercial yeasts |
Selected and controlled yeasts |
|
Pressing |
Manual or with rudimentary press |
Pneumatic or hydraulic presses |
|
Aging |
Young wine, no prolonged aging |
May have barrel (oak) and bottle aging |
|
Clarification and Filtration |
Minimal, sometimes unfiltered |
Controlled clarification and filtration |
|
Bottling |
Artisanal or semi-industrial |
Industrial, with quality controls |
|
Final Wine Style |
Light, aromatic (fragrant), low in tannins, marked acidity |
Structured, full-bodied, present tannins, aging potential |
|
Destination |
Local consumption, regional fairs, enological tourism |
Domestic and international export market |
Line 76 When the pomaces were dried, is there any inert gas protection?
Response. The inert gas protection was not used in both types of pomace.
Line 81 Appropriate references were needed for this method.
Response. A more detailed description of the determination process has been added (Lines 91-93).
Line 156 Is there any reference for the extraction process? Why is it different from the extraction method for Phenolic composition as describe line 131?
Response. We understand that it is not necessary to make a reference because this is an empirical technique, and we obtained the best extraction results under these conditions. Regarding the difference in the methanol used in each case, the HPLC analysis was carried out using the Nakov et al. technique, as this yields good results. This is why the extraction was different in each case, but this ensured that the extraction was the most appropriate for the analysis.
Line 220 The full name of the abbreviation shown in the figure 2 should be explained under the figure, such as Ash, TDF…
Response. The added meaning of the abbreviation is given in the figure legend.

Round 2
Reviewer 1 Report
Comments and Suggestions for Authors
Authors have addressed all the given comments and revised the manuscript accordingly; there are no further comments to offer. This paper can be accepted.
Author Response
Authors have addressed all the given comments and revised the manuscript accordingly; there are no further comments to offer. This paper can be accepted.
Response. Thank you very much for all your suggestions and recommendations, they have been very useful to improve the manuscript.
Reviewer 3 Report
Comments and Suggestions for Authors
Although previous study about the nutritional or antioxidants profiles of grape pomace were disscussed in the results, some crucial studies and findings also should be involved in the introduction, and it is very importment.
Author Response
Although previous study about the nutritional or antioxidants profiles of grape pomace were disscussed in the results, some crucial studies and findings also should be involved in the introduction, and it is very importment.
Response. Previous studies on the nutritional and antioxidant profiles of grape pomace were added in the introduction (Lines 49-52).